# Adalimumab Serum Concentrations, Clinical and Endoscopic Disease Activity in Crohn’s Disease: A Cross-Sectional Multicentric Latin American Study

**DOI:** 10.3390/pharmaceutics15020586

**Published:** 2023-02-09

**Authors:** Letícia Rodrigues de Souza, Daniela Oliveira Magro, Fábio Vieira Teixeira, Rogério Serafim Parra, Eron Fábio Miranda, Omar Féres, Rogério Saad-Hossne, Giedre Soares Prates Herrerias, Renato Mitsunori Nisihara, Claudio Saddy Rodrigues Coy, Ligia Yukie Sassaki, Paulo Gustavo Kotze

**Affiliations:** 1Colorectal Surgery Unit, Pontificia Universidade Católica do Paraná, PUCPR, Curitiba 80910-215, Brazil; 2Colorectal Surgery Unit, Universidade Estadual de Campinas, UNICAMP, Campinas 13083-970, Brazil; 3Clínica Gastrosaúde, Marília 17500-000, Brazil; 4Colorectal Surgery Unit, Universidade de São Paulo, USP, Ribeirão Preto 05508-090, Brazil; 5IBD Outpatient Clinics, São Paulo State University, UNESP, Botucatu 01049-010, Brazil; 6Department of Immunology, Mackenzie Evangelical School of Medicine, Curitiba 81280-330, Brazil

**Keywords:** adalimumab, Crohn’s disease, inflammatory bowel diseases, dosage, therapeutic drug monitoring

## Abstract

Despite some variability in ideal serum Adalimumab (ADA) concentrations, there is increasing evidence that higher concentrations of anti-TNF-α agents can be associated with sustained efficacy, and low or undetectable levels may lead to loss of response. This study aims to correlate serum ADA concentrations with clinical and endoscopic activity in patients with Crohn’s disease (CD). A cross-sectional and multicentric study was performed with patients with CD, who used ADA for at least 24 weeks. Patients were allocated into groups according to the presence of clinical or endoscopic disease activity. Serum ADA concentrations were measured and compared between groups. Overall, 89 patients were included. A total of 27 patients had clinically active CD and 62 were in clinical remission. Forty patients had endoscopic disease activity and 49 were in endoscopic remission. The mean serum ADA concentration was 10.2 μg/mL in patients with clinically active CD and 14.3 μg/mL in patients in clinical remission (*p* = 0.395). The mean serum ADA concentration in patients with endoscopic activity was 11.3 μg/mL as compared to 14.5 μg/mL in those with endoscopic remission (*p* = 0.566). There was no difference between serum ADA concentrations regarding clinical or endoscopic activity in CD, as compared to patients in remission

## 1. Introduction

Crohn’s disease (CD) is a chronic immune-mediated disease, with periods of exacerbation, resulting from an uncontrolled inflammation of the intestinal mucosa. Over two decades, much progress has been made in the treatment of CD. The approval of biological agents was an important step in this process. Biologics are effective in changing the natural history of CD, by healing the mucosa, preventing progression to severe forms, and reducing the rate of major abdominal surgery [1].

Adalimumab (ADA) is a fully human recombinant immunoglobulin G1 (IgG1) monoclonal antibody, administered subcutaneously, which binds with high affinity and specificity to soluble tumor necrosis factor (TNF) alpha. This anti-TNF-α agent is effective for the treatment of moderate-to-severe CD [1]. Its mechanism of action culminates in the reduction of T-cell proliferation and induction of apoptosis [1,2,3].

The efficacy of ADA was demonstrated in induction and maintenance pivotal studies, both in bio-naïve patients and in those previously exposed to infliximab. [3,4,5,6]. Despite TNF inhibitors comprised a landmark in the treatment of CD, these drugs can be associated with primary non-response (10–30%) or secondary loss of response (23–46%) after 1 year of treatment [7]. Biologic-related costs are significant, and as there are scarce effective therapeutic options in CD management, optimization of the use of each agent comprises a real need. Therapeutic Drug Monitoring (TDM) of anti-TNF-alpha agents involves the measurement of serum concentrations and anti-drug antibodies and has emerged as a strategy to guide treatment optimization and maximize benefits, by dose optimization or switching to different agents [7].

Therapeutic monitoring of ADA can represent an important tool to optimize therapy in patients with CD. Adequate serum concentrations of ADA are possibly associated with good clinical, biological and endoscopic outcomes. There is growing evidence that higher levels of anti-TNF-α can be related to a sustained response, and similarly, low or undetectable levels may increase the likelihood of loss of response [8,9,10,11,12].

Higher ADA concentrations were associated with disease remission (area under curve 0.748; *p* < 0.001) in a study by Van Hoeve et al. [10]. The identified cut-off point was 5.85 μg/mL, demonstrating good sensitivity and specificity and a similar association with remission predictors (68% and 70.6%, respectively). Serum ADA levels were inversely related to disease activity. Patients under ADA therapy with mucosal healing also had serum concentrations >6.5 μg/mL when compared to those who had partial healing (<4.0 μg/mL). Values between 8–12 μg/mL were shown to be adequate for mucosal healing in 80–90% of patients with CD [13,14]. Despite the lack of robustness of the data and the wide variability of ADA levels according to the clinical status of patients, the American Gastroenterology Association (AGA) recommends that the target serum ADA concentration to guide treatment is ≥7.5 μg/mL [15].

There are two assays to measure serum ADA concentrations. Performing an ELISA test requires a longer time and a need for multiple consecutive samples to reduce costs [16]. The rapid test, on contrary, has a reduced test time (approximately 15 min). The rapid test has been validated with good correlation with the Elisa assays (r2 = 0.90) and can represent a monitoring tool for ADA levels’ quantification in patients with CD. The Quantum Blue^®^ Adalimumab test measures levels ranging from 1.3 to 35 μg/mL, contemplating linear therapeutic values [7,14].

There are scarce data focusing on serum ADA levels in CD patients, particularly in the Latin American population. The aim of the present study was to correlate serum ADA concentrations with clinical and endoscopic activity in patients with CD. We hypothesized that patients in clinical or endoscopic remission would have significantly higher levels in comparison to those with active disease.

## 2. Materials and Methods

### 2.1. Study Design and Patient Population

This was a cross-sectional, observational, real-life, multicenter study performed in patients with CD treated at five tertiary centers from southern and southeastern Brazil. Consecutive patients from outpatient clinics from each center, under ADA treatment, could enter the study. Patients with a confirmed diagnosis of CD based on clinical, laboratory, endoscopic, radiologic and histopathological criteria for at least 6 months were included. Additional inclusion criteria comprised patients of any age who used ADA for at least 24 weeks, in doses of 40 mg every 2 weeks or weekly, who underwent the induction regimen with 160/80 mg, who agreed to participate in the study and signed the informed consent form. Patients with irregular use of ADA and patients with incomplete medical records were excluded.

### 2.2. Variables Analyzed

Demographic and clinical data such as age, sex, race, smoking status, body mass index (BMI), disease duration from diagnosis until blood sample collection, previous history of surgery, concomitant use of immunosuppressants, previous biologicals or corticosteroids were evaluated.

The Montreal Classification was used to assess the extent and behavior of CD [17]. Disease activity was evaluated using the Harvey-Bradshaw Index (HBI). Clinical remission was considered as HBI ≤ 4 points and clinical disease activity as a HBI > 4 points [18]. Endoscopic disease activity was assessed using the Simplified Endoscopic Score for Crohn’s Disease (SES-CD). The index is based on four endoscopic variables, including the size of the ulcers, the extent of the ulcerated surface, extension of affected area and the presence of stenosis. Each variable is rated on a score of 0 to 3 in each assessed segment. The score value ranges from 0–44. The higher the score, the greater the endoscopic severity of the disease. Endoscopic remission was defined as SES-CD score < 3 [19,20].

At the time of sample collection for the serum concentrations of ADA, biochemical tests such as: hematocrit, hemoglobin, erythrocyte sedimentation rate, albumin, and C-reactive protein, whenever available, were evaluated to assess disease activity. Fecal calprotectin levels were additionally quantified. The result was used as an additional parameter for the presence of inflammatory disease activity. Fecal calprotectin is a calcium-bound heterodimer with great abundance in the cytoplasm of neutrophils, which during the inflammatory process is released proportionally to the degree of inflammation [21]. This evaluation was performed using the Quantum Blue^®^ Calprotectin Extended method, LF-CALE (Buhlmann, Basel, Switzerland) available in commercial kits and following standards established by the supplier. Serum ADA was measured using the Quantum Blue^®^ Adalimumab test (Buhlmann, Basel, Switzerland), according to the manufacturer’s instructions. Blood samples were collected immediately before ADA injection. 

Patients were allocated into groups according to the presence of active CD or not (active disease or remission, clinical and endoscopic), according to previously described definitions. Serum ADA concentrations were measured and mean levels between the groups were compared.

### 2.3. Data Analysis

The variables were analyzed in terms of their mean and distribution pattern. Differences between the two groups were analyzed by parametric test (Student’s *t* test) for normally distributed variables. Qualitative variables were presented as percentages, and the chi-square test was used to compare two proportions (from independent samples). Fisher’s exact test was used for a small number of expected frequencies (when the total number of cases was less than 20), for which the chi-square test is not appropriate.

Boxplot graphics were used to provide the variability of serum ADA concentrations in active CD and remission. They show the median values, upper and lower quartiles, minimum and maximum values, and any possible outliers in the dataset. The significance level adopted for the statistical tests was 5%. The SPSS 16.0—Advanced Statistics software (IBM SPSS Statistics for Windows. IBM Corp; Armonk, NY, USA, 2013) was used for statistical analyses and graph editing.

### 2.4. Ethical Considerations

This study was centrally approved by the Research Ethics Committee from Sao Paulo State University (UNESP), in the ministry of health website, under reference number CAAE 88502318.2.1001.5411, and by ethical boards from each participating center. All patients signed a specific informed consent.

## 3. Results

Overall, 103 patients had ADA serum concentrations consecutively measured, with exclusion of 14 patients for having ulcerative colitis. Overall, data from 89 patients with CD were analyzed. A total of 27 patients (30.3%) had clinically active CD and 62 (69.7%) were in clinical remission. Forty patients (44.9%) had endoscopic disease activity and 49 (55.1%) were in endoscopic remission.

Table 1 describes in detail the baseline characteristics and demographics of patients between the 2 subgroups (according to the presence or not of clinically active CD), such as gender, BMI, smoking status, race, age, and treatment characteristics, among others. The CD phenotype was homogeneous between the groups, with the higher proportion of patients with no perianal disease, without previous biological treatment and without previous CD-related surgery. Regardless of the degree of disease activity, most of the population did not have anemia and had adequate albumin levels. As observed, three variables were significantly associated with the presence of clinically active CD: higher CRP (*p* < 0.05), higher Harvey-Bradshaw index (*p* < 0.05) and the presence of perianal disease (*p* = 0.011).

The mean serum concentration of ADA was 10.2 μg/mL (SD = 8.5; min–max values: <1.3–>35) in the clinically active group and 14.3 μg/mL (SD = 9.37; min–max values: <1.3–>35) in the remission group (*p* = 0.395). These results are illustrated in detail in the boxplot from Figure 1. Despite no significant difference, ADA serum levels were numerically higher in optimized (40 mg weekly) in comparison patients with standard dose (40 mg every 2 weeks), either in those with clinically active disease (10.5 μg/mL vs. 9.6 μg/mL; *p* = 0.501) and clinical remission (18.6 μg/mL vs. 13 μg/mL; *p* = 0.894).

The baseline characteristics of the subgroups in relation to the presence of endoscopic disease activity are illustrated in detail in Table 2. As observed, younger age (*p* = 0.029), higher CRP (*p* < 0.05), higher mean Harvey Bradshaw index (*p* < 0.05), disease behavior (*p* = 0.017) and the presence of perianal disease (*p* = 0.022) were associated with endoscopic activity. The mean serum ADA level in patients with endoscopic disease activity was 11.3 μg/mL (SD = 8.8; min–max values: <1.3–35) as compared to 14.5 μg/mL (SD = 9.49; min–max values: <1.3–>35) in the group with no endoscopic activity (*p* = 0.566). These data are illustrated in the boxplot from Figure 2. In regard of endoscopic activity, ADA serum levels in patients with optimized dose were numerically higher than patients with standard dose in the groups with endoscopic activity (11.4 μg/mL × 11.2 μg/mL; *p* = 0.386) and endoscopic remission (19.9 μg/mL × 13.1 μg/mL; *p* = 0.918), despite no significant statistical difference.

When analyzing different cut-offs to check clinical or endoscopic active CD, there was a significant association between clinical remission and ADA serum concentrations >5, >7.5 and >12 μg/mL. There was no significant association between the presence of endoscopic remission and the same cut-off points. These results are illustrated in detail in Table 3.

## 4. Discussion

Personalized treatment of CD is a current unmet need, as there is a lack of predictors to precisely identify the best agent with the best dosing for each individual patient. Strategies such as TDM and a “treat-to-target” approach aimed at mucosal healing have become the cornerstones of this form of treatment. Although recent evidence demonstrated that high serum concentrations of anti-TNF agents might be associated with disease remission, existing data in the literature are still conflicting [14]. In the present cross-sectional study, there was no association between higher serum ADA concentrations and clinical or endoscopic remission in patients with CD.

Mazor et al. conducted a study with 71 patients with CD treated with ADA. The authors concluded that there was an inverse relationship between elevated serum ADA concentrations and the presence of disease activity, with a serum concentration value of 5.85 μg/mL being the precise cut-off point to predict clinical remission [12]. Despite not establishing a significant association, our study showed numerically higher serum concentrations in patients in clinical remission in CD (10.2 vs. 14.3, *p* = 0.395). It is speculated if with a different methodology or with a larger sample of patients, this result could be significant.

Paul et al. conducted a meta-analysis with 14 studies involving 1914 patients with CD and ulcerative colitis [22]. Among these, only the study by Chiu et al. did not demonstrate a higher rate of clinical response in patients with higher serum ADA concentrations, not establishing an adequate cut-off serum level associated with clinical activity/remission in CD [23].

The ideal therapeutic range for ADA serum concentrations appears to be between 5 and 12 μg/mL, with some variability depending on the disease phenotype (fistulizing disease) or targeted treatment outcome (mucosal healing) [7,12,13,14,15]. Our data demonstrated that ADA levels > 5 μg/mL were associated with clinical remission. Yarur et al. described that ADA serum concentrations which may be necessary to achieve mucosal healing histologically and endoscopically are higher than those necessary to achieve clinical remission. In their study, there was a positive correlation between lower mean serum ADA concentrations and endoscopic activity [24]. Despite the low sensitivity, the cut-off level established was ≥7.5 μg/mL for the absence of endoscopic lesions. This value is recommended by the AGA to serve as a guide for treatment in CD patients under ADA treatment [15]. Our data did not demonstrate an association between levels > 7.5 μg/mL and endoscopic remission, probably due to the reduced sample of patients.

The bioavailability of anti-TNF agents is affected by several factors that can interfere with inter and intra-individual variability, including the use of immunosuppressants (decreases clearance), low albumin levels (increases clearance), high CRP levels (increased clearance), elevated baseline TNF levels (may increase clearance), sex (appears to increase clearance in men), development of anti-ADA antibodies (increase frequency of adverse events in addition to increased drug clearance), and high body mass index (may increase clearance) [25]. These are important variables which may interfere in serum ADA concentrations and need to be considered when interpreting the results of our study.

Some studies have demonstrated a significant association between serum levels of infliximab and better clinical and endoscopic outcomes in inflammatory bowel diseases (IBD). As it is an intravenous drug, with speculated more uniform pharmacokinetic and pharmacodynamic properties, treatment with infliximab may be better benefited by the TDM strategy. As ADA is a subcutaneous drug, its distribution can be more heterogeneous, which can make it difficult to interpret its results in relation to the dosage of its serum concentrations [26].

A recent prospective study (SERENE-CD) evaluated if higher doses of ADA would be associated to better clinical and endoscopic outcomes [27]. Patients with standard dosing (160/80 mg induction) had similar clinical remission rates at week 4 as compared to those with higher induction dose (4 weekly doses of 160 mg), 44% in both regimens (*p* = 0.939). No difference was also noted in endoscopic response at week 12 (39% vs. 43%, respectively; *p* = 0.463). Moreover, no differences were also observed among week 12 responders after 56 weeks. Despite the higher dosing regimen were associated to higher ADA serum concentrations, this was not reflected in differences in regard to clinical remission at week 4. Therefore, if higher ADA concentrations may lead to better clinical outcomes, this was not proved in this large prospective study. More research is warranted regarding TDM with ADA and its possible advantages in clinical practice.

Our results demonstrated that no differences in serum concentrations of ADA were identified neither in clinical nor in endoscopic remission, in comparison to active disease. However, when specific cut-off analyses were performed, there was a significant difference for clinical remission rates in ≥5, ≥7.5 and ≥12 µg/mL. Patients with higher concentrations than these pre-specified cut-offs presented higher proportions of clinical remission as compared to those with lower concentrations. The same was not observed in endoscopic remission in each different cut-off. We understand that endoscopic remission is a harder endpoint to be achieved as compared to clinical remission. Subjectivity of the HBI index in association to the reduced number of included patients in our sample may have influenced these results.

Lastly, another important issue in dosing ADA concentrations lies in the type of assay used. Comparisons between the Quantum rapid test and ELISA assays demonstrated that there can be discrepancy between these two methods. A Portuguese study demonstrated that the rapid test had intraclass correlation coefficients of 0.590, 0.864 and 0.761 when compared with 3 different ELISA assays [28]. Authors concluded that despite these differences, the rapid test is associated with the advantage of having results in 15 min, and can be considered as an important tool for gastroenterologists in the management of patients treated with ADA. Laserna-Mendieta et al. evaluated the interchangeability and agreement between the Quantum method and two established ELISA kits, Promonitor and Lisa-Tracker [29]. Statistical differences were identified between these methods, with the rapid test possibly overestimating values of ADA concentrations. Greater differences were identified in higher concentrations, and in subtherapeutic levels, the correlation with ELISA methods was high. In our cohort, the Quantum rapid test was used in all patients due to better access to this method in our country. Despite the rapid test and ELISA assays cannot be interchangeable, the use of the rapid test represents a valuable alternative in TDM with ADA in daily practice when used as the single method, multiple times, in the same patient.

The present study is associated with some limitations, which must be considered in the final analysis of the results. The small number of participants may have contributed to the lack of possible associations of variables, in contrary with most of the previous evidence. Most of our patients were in clinical remission. Another limitation was the scarcity of some data on some of the variables analyzed, as not all patients had complete data (mostly laboratory parameters) at the time of blood sample collection of drug concentrations. In addition, the measurement of ADA concentrations did not follow a proactive or reactive strategy, being performed with a cross-sectional methodology, with a convenience sample (no powered sample calculation). Another important issue is the absence of comparison with other ELISA assays. Despite these limitations, the study has strengths that deserve to be highlighted. The cross-sectional design helped to reduce data collection biases. The multicentric feature of the study, performed in tertiary referral centers, contributed to reduce bias. In addition, this is the first study with serum concentrations of ADA in CD patients from Latin America, which can act as stimulation for further research with strategies to optimal use of biologics in our continent.

In summary, there were no differences between serum ADA concentrations regarding the presence of clinical or endoscopic disease activity in CD. A trend towards higher serum levels was observed in patients in clinical remission, without statistical significance. ADA levels > 5 μg/mL were associated to clinical but not endoscopic remission. Further studies are warranted to better position TDM with ADA in the management of CD.

## Figures and Tables

**Figure 1 pharmaceutics-15-00586-f001:**
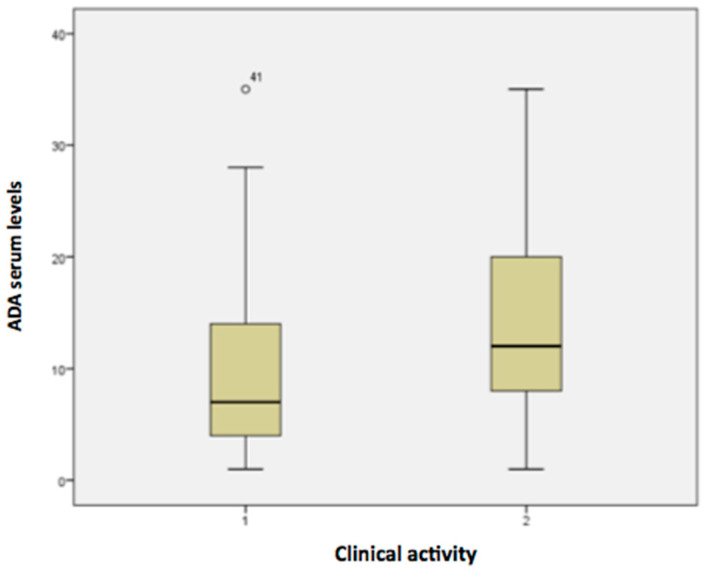
Boxplot demonstrating median serum ada concentrations and clinical disease activity in cd (1: active disease; 2: clinical remission).

**Figure 2 pharmaceutics-15-00586-f002:**
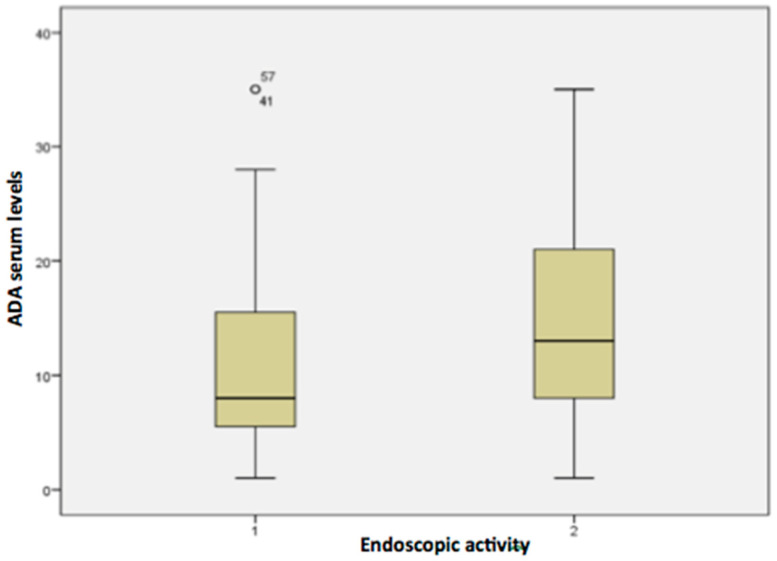
Boxplot demonstrating median serum ADA concentrations and endoscopic disease activity in cd (1: endoscopically active disease; 2: endoscopic remission).

**Table 1 pharmaceutics-15-00586-t001:** Baseline characteristics of patient population analyzed in subgroups regarding clinical disease activity in cd. Χ^2^ chi-square; *t* test.

	Active Disease (n = 27)		Clinical Remission (n = 62)		*p*-Value
**Patients (N = 89)**					
Age (years) (N = 89)	39 ± 14.8		45 ± 13.7		0.518
Duration of the disease (months) (N = 89)	112.6 ± 100.7		132.5 ± 104.5		0.874
Optimization time (months) (N = 31)	12.2 ± 9.1		17.9 ± 13.9		0.194
CRP (N = 77)	8.2 ± 8.2		1.8 ± 3.4		<0.05
ESR (N = 49)	35.6 ± 17.1		17.3 ± 20.8		0.705
Albumin (N = 50)	4.1 ± 0.46		4.1 ± 0.30		0.208
Hemoglobin (N = 76)	12.9 ± 1.9		13.2 ± 1.8		0.977
Hematocrit (N = 76)	38.7 ± 4.9		39.9 ± 4.6		0.624
Calprotectin (N = 63)	516 ± 397.9		231.9 ± 378.3		0.304
ADA serum level (N = 89)	10.2 ± 8.5		14.3 ± 9.4		0.395
ADA levels in optimized (Md ± SD)	10.5 ± 9.5	*p* = 0.501	18.6 ± 8.9	*p* = 0.894	
ADA levels in non-optimized (Md ± SD)	9.6 ± 6.8		13 ± 9.2		
BMI (N = 89)	24.1 ± 4.9		25.9 ± 5.2		0.854
HBI (N = 89)	8.6 ± 3		1.6 ± 1.5		<0.05
Female (%) (N = 89)	16 (17.9)		37 (41.6)		0.971
Caucasian (%) (N = 89)	22 (24.7)		50 (56.2)		0.472
Non-smokers (%) (N = 89)	21(23.6)		46 (51.7)		0.821
*Montreal Classification*					
Age at onset—N (%) (N = 89)					0.134
A1: ≤16 years	4 (4.5)		2 (2.2)		
A2: 16–40 years	17 (19.1)		44 (49.4)		
A3: >40 years	6 (6.7)		16 (17.9)		
Disease location—N (%) (N = 89)					0.703
L1: terminal ileum	7 (1.1)		16 (17.9)		
L2: colonic	3 (3.4)		10 (11.2)		
L3: ileocolonic	17 (19.1)		34 (38.2)		
L4: isolated upper GI disease	0		2 (2.2)		
Disease behaviour—N (%) (N = 89)					0.145
B1: nonstricturing, nonpenetrating	13 (14.6)		17 (19.1)		
B2: stricturing	8 (8.9)		22 (24.7)		
B3: penetrating	6 (6.7)		23 (25.8)		
Perianal disease (%) (N = 89)	16 (17.9)		19 (21.3)		0.011
Use of imunossupressants					
AZA (%) (N = 89)	16 (17.9)		25 (28)		0.099
MTX (%) (N = 89)	1 (1.1)		0		0.128
Corticosteroids (%) (N = 89)	13 (14.6)		18 (20.2)		0.082
Previous use of biologicals (%) (N = 89)	10 (11.2)		16 (17.9)		0.284
Previous surgery (%) (N = 89)	16 (17.9)		38 (42.7)		0.857

**Table 2 pharmaceutics-15-00586-t002:** Baseline characteristics of patient population analyzed in subgroups regarding endoscopic disease activity in cd. Χ^2^ chi-square; *t* test.

	Endoscopic Activity (n = 40)		Endoscopic Remission (n = 49)		*p*-Value
**Patients (N = 89)**					
Age (years) (N = 89)	39 ± 16.1		46 ± 11.7		0.029
Disease duration (months) (N = 89)	83.1 ± 78.2		161.9 ± 108.3		0.072
Optimization time (months) (N = 31)	14.4 ± 9.4		15.7 ± 15.9		0.178
CRP (N = 77)	7.4 ± 7.7		1.1 ± 2.1		<0.05
ESR (N = 49)	28.7 ± 21.2		18.3 ± 20.6		0.358
Albumin (N = 50)	4 ± 0.36		4.2 ± 0.36		0.299
Hemoglobin (N = 77)	12.7 ± 1.8		13.4 ± 1.7		0.942
Hematocrit (N = 77)	38.4 ± 4.7		40.4 ± 4.5		0.466
Calprotectin (N = 63)	397.1 ± 373		245.24 ± 421.2		0.601
ADA serum levels (N = 89)	11.3 ± 8.8		14.5 ± 9.5		0.566
ADA levels in optimized (Md ± SD)	11.4 ± 9.4	*p* = 0.386	19.9 ± 9.1	*p* = 0.918	
ADA levels in non optimized (Md ± SD)	11.2 ± 8.3		13.1 ± 9.2		
BMI (N = 89)	24.5 ± 4.9		26.2 ± 5.3		0.513
HBI (N = 89)	5.8 ± 4.3		2 ± 2.4		<0.05
ADA time of use (months) (N = 89)	46.1 ± 32.7		55.6 ± 34.9		0.777
Female (%) (N = 89)	24 (26.9)		29 (32.6)		0.938
Caucasian (%) (N = 89)	34 (38.2)		38 (42.7)		0.269
Non-smokers (%) (N = 89)	31 (34.8)		36 (40.4)		0.874
*Montreal Classification*					
Age at onset—N (%) (N = 89)					0.102
A1: ≤16 years	5 (5.6)		1 (1.1)		
A2: 16–40 years	24 (26.9)		37 (41.6)		
A3: >40 years	11 (12.4)		11 (12.4)		
Disease location—N (%) (N = 89)					0.619
L1: terminal ileum	13 (14.6)		10 (11.2)		
L2: colonic	5 (5.6)		8 (8.9)		
L3: ileocolonic	21 (23.6)		30 (33.7)		
L4: isolated upper disease	1 (1.1)		1 (1.1)		
Disease behaviour—N (%) (N = 89)					0.017
B1: nonstricturing, nonpenetrating	19 (21.3)		11 (12.4)		
B2: stricturing	8 (8.9)		22 (24.7)		
B3: penetrating	13 (14.6)		16 (17.9)		
Perianal disease (%) (N = 89)	21 (23.6)		14 (15.7)		0.022
Use of imunossupressants					
AZA (%) (N = 89)	22 (24.7)		19 (21.3)		0.127
MTX (%) (N = 89)	1 (1.1)		0		0.266
Corticosteroids (%) (N = 89)	18 (20.2)		13 (14.6)		0.069
Previous use of biologicals (%) (N = 89)	14 (15.7)		12 (13.5)		0.278
Previous surgery (%) (N = 89)	23 (25.8)		31 (34.8)		0.58

**Table 3 pharmaceutics-15-00586-t003:** Clinical and endoscopic disease activity according to different cut-off values of ADA serum concentrations.

Cut-Off of Serum ADA Concentrations (µg/mL)						
	Cut-Off 5.0 µg/mL		Cut-Off 7.5 µg/mL		Cut-Off 12 µg/mL	
	<5	≥5	<7.5	≥7.5	<12	≥12
Clinical disease activity, n (%)						
Yes	9 (50)	19 (25)	15 (50)	12 (20.3)	17 (37.0)	10 (23.3)
No	9 (50)	57 (75)	15 (50)	47 (79.7)	29 (63.0)	33 (76.7)
*p*-value	0.573	0.0001	1.000	0.0001	0.109	0.001
Endoscopic disease activity, n (%)						
Yes	8 (61.5)	31 (40.8)	18 (60)	22 (37.3)	25 (54.3)	15 (34.9)
No	5 (38.5)	45 (59.2)	12 (40)	37 (62.7)	21 (45.6)	28 (65.1)
*p*-value	0.255	0.139	0.362	0.067	0.662	0.069

## Data Availability

Full data is available under request, once approved by central IRB.

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
