# Peer review of "Adalimumab Serum Concentrations, Clinical and Endoscopic Disease Activity in Crohn’s Disease: A Cross-Sectional Multicentric Latin American Study"

_pharmaceutics, 2023, doi:10.3390/pharmaceutics15020586_

Round 1

Reviewer 1 Report

The aim of the study was to investigate the correlation between adalimumab trough levels with clinical and endoscopic remission in patients with Crohn's disease who had had treatment for at least six months. Of 89 patients studied, there was no difference in serum levels depending on whether the patients were in clinical or endoscopic remission or not. The Quantum test used for the residual levels is a rapid, point-of-care test, which is not Elisa but a chromo-immunology technique.

Comments/questions:

1. is it possible to give the doses of adalimumab in the results (optimized versus non-optimized patients) in tables 1 and 2?

2. What is the definition of clinical remission?

3. What is the definition of endoscopic remission?

4. the authors do not report any study of the pharmacokinetics of the evolution of the levels with this Quantum technique compared to the reference technique Elisa and it seems to me that there is a differential of at least 15% difference between the results of the two techniques with however a relatively parallel linearity (data internal to the service, not published).

General comment:

The whole study is based on a comparison with trough levels collected by the Quantum technique, which is not the reference technique. The article can be published if the authors report the data from the literature on the Quantum technique and its comparison with the results of the reference technique in Elisa. There must be a discussion on the difference between the two techniques and why this new technique has not also become a reference technique in daily practice.

Author Response

Yours Sincerely.

Reviewer 2 Report

I have the following major remarks:

1. The discussion on the discrepancy in the demonstrated trends of ADA correlation with endoscopic and clinical CD activity would be interesting as we know that endoscopic remission is the main goal of the therapy and it is not a necessary association between clinical and endoscopic activity.

2. The title should be modified to reflect the geographic area of the study as it is a strength of the project.

I have the following minor remarks:

1. The paper needs language polishing as well as editing. For instance, in table 1, line 193, the title of the table should be in the capital; 

2. Legends of the tables and figures are missing.

3. Line 321 - sentence "The search for consensus on the use of the strategy of monitoring serum concentrations of ADA in IBD still 321 persists due to different findings in the international literature" it is not a conclusion from the study.

4. Line 316 -  the second part of the sentence is not in connection with the first part as well as with the study itself "In addition, this is the first study with 315 serum concentrations of ADA in CD patients from Latin America, which demonstrates that IBD care in the continent 316 is improving, in line with recent recommended international tendencies and strategies".

Author Response

Yours Sincerely

Round 2

Reviewer 1 Report

OK for this reviewed version. I would suggest a different title:

"Adalimumab  serum  concentrations,  clinical  and  endoscopic disease  activity  in  Crohn's  Disease:  a  cross-sectional multicentric Latin American study with a point of care non Elisa test"